# From Soil to Grape and Wine: Geographical Variations in Elemental Profiles in Different Chinese Regions

**DOI:** 10.3390/foods10123108

**Published:** 2021-12-15

**Authors:** Xiaoyun Hao, Feifei Gao, Hao Wu, Yangbo Song, Liang Zhang, Hua Li, Hua Wang

**Affiliations:** 1College of Enology, Northwest A&F University, Yangling, Xianyang 712100, China; haoxiaoyun@nwafu.edu.cn (X.H.); gaofeifei@nwafu.edu.cn (F.G.); zhangliang20@nwafu.edu.cn (L.Z.); lihuawine@nwafu.edu.cn (H.L.); 2Food Inspection and Quarantine Center, Shenzhen Customs, Shenzhen 518033, China; whakyo@gmail.com; 3Agriculture and Animal Husbandry College, Qinghai University, Xining 810015, China; ys792@cornell.edu; 4Shaanxi Engineering Research Center for Viti-Viniculture, Yangling, Xianyang 712100, China; 5Engineering Research Center for Viti-Viniculture, National Forestry and Grassland Administration, Yangling, Xianyang 712100, China; 6China Wine Industry Technology Institute, Yinchuan 750021, China

**Keywords:** geographical variation, elemental profile, bioconcentration factor, transfer factor, environmental factor, ICP-MS

## Abstract

Elemental profiles are frequently applied to identify the geographical origin and authenticity of food products, to guarantee quality. The concentrations of fifteen major, minor, and trace elements (Na, Mg, K, Ca, Al, Fe, Mn, Cu, Zn, Rb, Sr, Li, Cd, Cs, and Ba) were determined in soils, “Meili” grapes, and wines from six regions in China by inductively coupled plasma mass spectrometry (ICP-MS). The elemental concentrations in these samples, according to the geographical origins, were analyzed by one-way analysis of variance (ANOVA) with Duncan’s multiple comparisons. The bioconcentration factor (BCF) from soil to grape and the transfer factor (TF) from grape to wine were calculated. Mg, K, Ca, Cu, Zn, Rb, Sr, and Ba presented higher BCF values than the other seven elements. The TF values of six elements (Na, Mg, K, Zn, Li, and Cs) were found to be greater than one. Moreover, the correlation of element content between the pairs of soil–grape, grape–wine, and bioconcentration factor (BCF)–environmental factor were analyzed. Significant correspondences among soil, grape, and wine were observed for K and Li. Two elements (Sr and Li) showed significant correlations between BCF and environmental factor (relative humidity, temperature, and latitude). A linear discriminant analysis (LDA) with three variables (K, Sr, Li) revealed a high accuracy (>90%) to determine the geographical origin for different Chinese regions.

## 1. Introduction

The geographical origins of food products provide a guarantee for quality, authenticity, and type [1]. The origin of wine, as a natural product, is an essential basis for the quality characteristics of wine products; it determines the characteristics of wine and plays a decisive role in the choices of consumers [2]. Hence, establishing the geographical origin of wine has become a major concern for countries worldwide [3].

After a long period of significant expansion, global vineyard areas and grape production has increased slowly, and wine consumption showed a 3% decrease compared to 2019 due to the COVID-19 crisis [4]. Concerning China, the growth of vineyards (758 kha) increased by 0.6% in 2020 compared to 2019. The consumption of wine (12.4 Mhl) in 2020 showed a 17.4% drop in relation to 2019. However, China still occupies the third highest position in terms of vineyard areas globally, and it is the sixth highest wine-consuming country [4]. Further, with the sharp decline in imported wine, the demand for domestic wine has increased, and “origin” has become a viable, value-added processing option. As the wine industry expands into new provinces, such as Shaanxi, Shanxi, and Inner Mongolia, so does the demand for geographical origin identification in China.

At present, the main techniques that have been implemented for wine origin identification include spectrum (near-infrared, middle-infrared, and nuclear magnetic resonance) [5,6,7], chromatography (gas chromatograph and liquid chromatography) [8,9,10], and mass spectrometry [11,12]. Mass spectrometry (MS), such as inductively coupled plasma mass spectrometry (ICP-MS) and isotope ratio mass spectrometry (IRMS), has become a frontline technology, rapidly replacing other methods in many fields of food science. In the field of food authentication, MS techniques have several advantages, such as high sensitivity, selectivity, throughput, and multi-analyte capabilities [13,14,15,16]. Therefore, numerous studies have been dedicated to the application of MS analysis in the provenance of wine [12,17,18,19]. Unlike the analysis of isotopes with IRMS, multielement analysis with ICP-MS is fast, inexpensive, and extensive. As a complex matrix, wine is rich in elements (major, trace, and rare earth elements) [20]. The primary components of wine are mainly derived from grapes, by absorbing them from soils in which the vines are grown. Moreover, anthropogenic factors, such as viticulture practices and vinification processes, can also contribute to elemental profiles [20]. Among the factors mentioned above, the geochemistry of vineyard soil plays a fundamental role in the elemental uptake of grapevines [16].

Consequently, several countries have successfully identified geographical origins by elemental analysis, such as Italy [18], Greece [21], Japan [22], United States [23], South Africa [24], Australia [25], Romania [26], and Slovenia [27]. Furthermore, these studies cite the superiority of the ICP-MS technique for determining elements. Simultaneously, multivariate statistical analyses, such as principal component (PCA), linear discriminant (LDA), and cluster analysis (CA) have been widely employed in the classification and pattern modeling of wine regions [6,11,12,17,21,22]. Generally, most of the studies regarding wine origin identification are conducted based on the analysis of elemental concentrations in wine combined with chemometric methods. Conversely, only a few studies have evaluated elemental migration and bioaccumulation in the soil–grape–wine system. Furthermore, particularly for Chinese wine regions, there are scarce published data on the elemental profiles in soils, grape berries, and wines [28,29,30,31]. To our knowledge, there are no studies combining bioconcentration factors (BCF), transfer factors (TF), and environmental conditions (precipitation, temperature, average relative humidity, latitude, and altitude) with elemental profiles, in order to obtain the characteristic elements for accurate wine classification in China.

In our study, “Meili” (*Vitis vinifera* L.), an innovative cultivar with strong disease resistance, bred by the Enology College, Northwest A&F University From 1982 to 1999, was chosen as the research material [32]. This cultivar was selected to resist disease (*Plasmopara viticola* and *Sphaceloma ampelinum*) and cold weather [32]. After years of cultivation experiments in Shaanxi, Shanxi, Inner Mongolia, and other areas, this variety showed strong disease resistance, a stable quality; it is a medium variety when it comes to producing distinct and flavorful red wine [33,34,35]. Recent studies have revealed that the uptake of major and trace elements could be affected by rootstock types [18,36,37]. While Pisciotta et al. [38] observed that different grafting combinations were not able to induce significant differences in the rare earth elements of Y, La, and lanthanides, their abundance in soils is scattered and need to be normalized. Our study object, the cultivar of “Meili” (*Vitis vinifera* L.), is own-rooted with a purity line named “8804”, which avoids the interference of rootstocks on major and trace elemental concentrations [32].

Therefore, comparisons among elements in soils, grapes, and wines from six regions were performed to explore the possible distinct composition in different regions. The relationships between the two pairs of soil–grape and grape–wine were considered to obtain geographical variations in elements. Meanwhile, to reveal the migration of elements in the soil–grape–wine system, the BCF values of soil–grape and the TF of wine–grape were calculated. Further, the relationships between BCF and environmental factors (temperature, precipitation, average relative humidity, altitude, and latitude) were investigated to obtain elemental profiles closely related to geographical origins. LDA and PCA was employed to confirm the geochemical markers of Chinese wine.

## 2. Materials and Methods

### 2.1. Sample Collection

Soil, grape, and wine samples were collected from three provinces (Shaanxi, Shanxi and Henan) and the Inner Mongolia autonomous region. These provinces are distinctive viticultural regions in China. Samples (soil, grape, and wine) were collected from six wine regions in the 2018 harvest year: YL (Yangling, Shaanxi), BLY (Bailuyuan, Shaanxi), HY (Heyang, Shaanxi), XX (Xiaxian, Shanxi), MQ (Minquan, Henan), and WH (Wuhai, Inner Mongolia Autonomous Region) regions (Figure 1). Table 1 shows the information of all samples, including geographical and climatic factors during veraison (July) and maturation periods (August).

The study is focused on wines made through the micro-vinification of “Meili” mono-cultivar. One hundred grape berries were picked from twenty bunches at random for each sampling plot and five grape samples for each region. Additionally, 25 kg of grape samples was collected from each region for winemaking. The plastic containers used for storing and treating the samples were cleaned to avoid contamination. The grape berries were stored at −20 °C until analysis. Furthermore, the 25 kg of grape samples for winemaking was destemmed and crushed into glass fermenters immediately after being taken back to the laboratory. 

Soil samples (*n* = 30) were collected from the corresponding vineyards. To reduce the effect of surface contamination (pollution and fertilizer), five soil samples (1.5 kg each) were collected from a depth of 10 to 60 cm, about 30 cm around each grapevine in the vineyard. In the laboratory, soil samples were dried to a constant weight and homogenized. Homogenization was performed by quartering and pulverization procedures.

### 2.2. Sample Pretreatment

Before elemental analysis, the samples were carefully prepared to avoid chemical and physical interactions. All tests were carried out at the Food Inspection and Quarantine Center of Shenzhen Customs in China.

#### 2.2.1. Fermentation Process

The details of micro-vinification procedures were performed as follows: grape clusters were destemmed and crushed into 5 L glass fermenters at the laboratory of Enology College, Northwest A&F University. After a three-day cold maceration at a temperature less than 10 °C, the must was warmed to 21 °C for natural fermentation using wild *Saccharomyces cerevisiae* from the grape skin itself. After two days of fermentation at 21 °C, the fermentation temperature was allowed to increase to 28 °C until the alcoholic fermentation was complete. No malolactic fermentation was performed. Then, molecular SO_2_ was adjusted to 0.6 mg/L with potassium metabisulfite (Food Grade, China).

#### 2.2.2. Digestion Process

##### Grape

Grapes were rinsed with Milli-Q water (resistivity 18.2 MΩ/cm, Milli-Q Advantage A10, Burlington, MA, USA) and blotted with clean paper. Then, 100 berries were homogenized using a homogenizer (JYL-C010 Jiuyang, Jinan, China), and acid digested using a microwave system (Ethos One, Milestone, Milan, Italy) with 75 mL Teflon digestion vessels. Next, 1 g homogenized samples were weighed inside, and 5 ml concentrated nitric acid (GR, Merck, Germany) was added. Teflon digestion vessels were pre-cleaned in 10% nitric acid solution (GR, Merck, Darmstadt, Germany) to avoid cross-contamination. Then, the vessels were transferred into the microwave digestion instrument for digestion by application of a program described by Geana et al. [26]. After cooling to ambient temperature, the content was quantitatively transferred into a 50 mL volumetric flask and brought to the volume with Milli-Q water. A blank calibration sample was made of HNO_3_; water and a quality control sample were also prepared using the above procedure.

##### Wine

The wine digestion was carried out in a Teflon digestion vessel, adding 2 mL of wine and 5 mL of HNO_3_. The vessels that were already capped were placed in a microwave oven, followed by applying the program described by Geana et al. [26]. The cooled digests were transferred into a 50 mL volumetric flask. The digestion tubes were rinsed three times with Milli-Q water, and this water was transferred into the volumetric flask and diluted with Milli-Q water up to 50 mL. 

##### Soil

Soil samples were dried at 105 °C for 24 h, then homogenized and passed through a 20-mesh sieve, after grinding, passed through an 80-mesh sieve to obtain very fine particles. An amount of 0.25 g of powder was weighted in a 50 mL Teflon digestion vessel, adding 3 mL of 65% HNO_3_, 2 mL concentrated HCl (GR, Merck, Darmstadt, Germany), and 1 mL concentrated HF (GR, Merck, Darmstadt, Germany). The digestion was carried out with the program by Geana et al. [26]. The final digests were transferred to volumetric flasks, and were filled up to 50 mL with Milli-Q water. 

### 2.3. Elemental Analysis

Elemental determination was performed with an inductively coupled plasma mass spectrometer (ICP-MS) (ICAP-Q, Thermo Fisher, Waltham, MA, USA) equipped with an ASX-260 auto-sampler (CETAC Thermo Fisher, Waltham, MA, USA), a Burgener nebulizer (1.0 mL/min), nickel cones, and a peristaltic sample delivery pump. For grape and wine samples, the operating conditions are described by Chinese standard [39]. For soil samples, kinetic energy discrimination (KED) mode was employed to minimize the interference of the instrument. The working parameters of ICP-MS was described by Wang et al. [40].

The quantified concentrations of elements were performed using an external standard of a calibration mixture standard solution (CLMS-2, SPEX CertiPrep, Metuchen, NJ, USA) in 0.5% nitric acid. The calibration range for major and trace elements was from 0.1 to 10 mg/L and from 0.001 to 1 mg/L, respectively. Rh and In (50 μg/L) were included as an internal standard to control instrumental drift correction. The multi-elemental standard solution of 200 mg/L for Ca, K, Mg, Na, Fe, and Mn, and 50 mg/L for other trace metals was used for a standard experimental recovery check. The recovery rate should be maintained at 90–110%. If the recovery exceeds the tolerance range, the batch of samples should be remeasured.

### 2.4. Statistical Analysis

The bioconcentration factor (BCF) was used to assess the transfer of elements from soils to grape berries, calculated using the equation below:BCF = C_g_/C_s_
(1)
where C_g_ and C_s_ are the concentrations of the elements in grape berries and soils, respectively.

The transfer factors (TF) were calculated to assess traceability and bioavailability of the elements from grape berries to wines.
TF = C_w_/C_g_(2)

The C_w_ and C_g_ represent the concentrations of the elements in wines and grape berries, respectively.

One-way analysis of variance (ANOVA) with Duncan’s multiple comparisons were applied to investigate whether the differences in the elemental concentrations in soil, grape, wine samples, BCF, and TF were relative to the six origins. Differences among the regions were considered significant when *p* < 0.05. The principal component analysis (PCA) was carried out to estimate elemental composition variables among different regions in a data matrix by a low-dimensional model. Pearson correlation analysis was carried out to confirm the relationship between the two pairs of soil–grape and grape–wine for elemental concentrations. Furthermore, the relationships between environmental factors (temperature, precipitation, average relative humidity, altitude, and latitude) and BCF values were analyzed to obtain some influences on the element uptake of the grape berries from the soils. Linear discriminant analysis (LDA) was utilized to evaluate whether the samples could be classified amongst geographical origins according to F-value. The robustness of the classification model was evaluated using a cross-validation test based on the “leave-one-out” procedure. All procedures were performed using SPSS 20.0 software (IBM Inc., Armonk, NY, USA).

## 3. Results

### 3.1. Elemental Compositions

#### 3.1.1. Elemental Concentrations in Soils

The mean elemental concentrations and standard deviations for soil samples collected from six regions are summarized in Table 2. Regarding the differences in the six studied regions, the fifteen elements determined all showed significant differences (*p* < 0.05), demonstrating that soils from different regions had a characteristic elemental profile. Among the major elements, Fe was the most concentrated, followed by K, Ca, Al, Na, and Mg. Fe presented the highest values in YL and BLY soils, followed by XX and HY; the lowest values were found in MQ and WH. With regard to the minor and trace elements, the concentrations order was Mn > Sr > Zn > Rb > Li > Cu > Ba > Cs > Cd, except for an outlier of Ba (mean value 330 mg/kg) in BLY. The concentration of Mn displayed the highest value in BLY, followed by YL, XX, and HY; the lowest values were found in WH and MQ regions. In general, most elements presented the highest values in YL, BLY, and XX; intermediate values were found in HY; the lowest values were in the WH and MQ regions.

#### 3.1.2. Elemental Concentrations in Grapes

Table 3 displays the mean elemental concentrations and standard deviations of grapes. Ten elements (Na, Mg, K, Mn, Rb, Sr, Li, Cd, Cs, and Ba) showed significant differences among the six regions (*p* < 0.05). The concentration order of major elements was K > Mg > Na. K showed the highest values in the XX region, with intermediate values in YL, BLY, MQ, and WH regions; the lowest values were found in the HY region. Concerning minor and trace elements, their concentration decreased in the order Rb > Sr > Mn > Ba > Li > Cs > Cd. The concentration ranges of Rb and Sr varied greatly among the six regions; their mean values were from 1.0 mg/kg (WH) to 7.3 mg/kg (BLY) and from 0.4 mg/kg in BLY to 7.0 mg/kg in XX, respectively. Overall, most elements in XX presented the highest values, while the lowest values were found in the WH region.

#### 3.1.3. Elemental Concentrations in Wines

The comparisons of elemental concentrations performed by grouping the wines from six regions are summarized in Table 4. These comparisons revealed significant differences for fifteen elements among the six regions (*p* < 0.05). As observed, K was the most concentrated element among the major elements, followed by Mg and Ca, while Na showed the lowest concentration. With regard to minor and trace elements, the contents in the studied wines decreased in the order Rb > Fe > Zn > Sr > Mn > Al > Cu > Ba > Li > Cs > Cd. The patterns of elemental concentrations in wines were the same as in the grapes. Moreover, the concentrations of Mn, Cu, Zn, Rb, Sr, Li, Cs, and Ba presented distinctive values among the six studied areas (*p* < 0.05). The wine samples from XX showed the highest concentrations of Mn, Zn, and Sr. The lowest values of elements (Fe, Zn, Sr, Li, and Ba) were found in the BLY region and the concentrations of Mn, Rb, Cd, and Cs displayed the lowest values in WH. 

### 3.2. Relationships among Elemental Concentrations between Two Pairs, Soil–Grape, and Grape–Wine

#### 3.2.1. Correlation of Elemental Concentrations between Soil and Grape

In order to identify the correlation of elemental concentrations between soil–grape and grape–wine, the Pearson correlation analysis and linear curve fitting were performed. According to the correlation coefficients and linear curve fitting, three elements show significant correlations between soil and grape berries (Figure 2). Two elements (K, Cd) present positive correlations (both of their r = 0.66, *p* < 0.01) (Figure 2a,b), while Li shows negative correlations (r = −0.74, *p* < 0.01) between soils and grape berries (Figure 2c).

#### 3.2.2. Correlation of Elemental Concentrations between Grape and Wine

Figure 3 presents positive relationships between grape berries and wines. Of the 15 elements, 6 elements (K, Mn, Rb, Sr, Li, and Cs) show a positive correlation between grape berries and wines (*p* < 0.01).

### 3.3. Bioconcentration Factor and Transfer Factor

#### 3.3.1. Bioconcentration Factor in Different Regions

According to the one-way ANOVA and Duncan’s test, the concentrations of all fifteen elements in the soil were significantly different (*p* < 0.05), ten elements in the grape samples showed significant differences (*p* < 0.05). As shown in Appendix A, the ability of grapes to absorb different elements varied greatly. After assessing the elemental bioconcentration, the values of the BCF for most elements investigated were lower than 1. The ranges of BCF values were the following: >1 for Rb in BLY region; 0.01–0.47 for Mg, K, Ca, Cu, Zn, Rb, Sr, and Ba; <0.01 for Na, Al, Fe, Mn, Li, Cd, and Cs.

#### 3.3.2. Relationships among the BCF Values

Pearson correlation analysis of the BCF values is shown in Appendix A. Among the fifteen BCF values, Rb and Mg showed the strongest correlation with the other eleven elements. The BCF values of Rb showed significant positive correlations with K, Cs, and Cd (*p* < 0.05), while demonstrating significant negative correlations with Na, Mg, Mn, Fe, Li, and Ba, as well as Cu and Sr (*p* < 0.05). The BCF values of Mg were significantly positively correlated with eight elements (Ca, Mn, Fe, Zn, Ba, Cu, Sr, and Li; *p* < 0.05), whereas significant negative correlations were presented with Rb, Cd, and Cs (*p* < 0.05).

#### 3.3.3. Relationships between BCF Values and Environmental Factors

The input and output of elements in the soil–plant system were affected by temperature, humidity, precipitation, and other meteorological factors [41,42]. As shown in Figure 4, the BCF values of Sr and Li were significantly correlated with environmental factors (*p* < 0.05). The BCF values of Sr presented a significant negative correlation with the average relative humidity (Figure 4a, r = −0.69, *p* < 0.05). The BCF values of Li element displayed a negative correlation (Figure 4b, r = −0.79, *p* < 0.05) with temperature. Moreover, the BCF value of Li was positively correlated (Figure 4c, r = 0.79, *p* < 0.05) with latitude.

#### 3.3.4. Transfer Factor in DIFFERENT regions

The transfer factor (TF) values of the wine-grape system are shown in Appendix A. The TF values of Na and Mg were >1 from all the studied regions. Four elements (K, Zn, Li, and Cs) presented for >1 TF values in part of regions. The ranges of TF values for Ca, Al, Fe, Mn, Cu, Rb, Sr, Cd, and Ba were all below 1.

### 3.4. Principal Component Analysis

PCA was performed with fifteen elements to visualize the sample distribution according to the geographical origin. As shown in Figure 5, the contributions of the first two principal components (PCs) explained 43.1% and 27.1% of the variance, comprising 70.2% of the total variance for soil samples. The first PC was strongly associated with the values of Mn, Cu, Fe, and Li, whereas Cs, Cd, Sr, and Zn were the dominant variables in the second PC (Figure 5a). For grape samples, the first two PCs (PC1, 29.6%; PC2, 20.9%) accounted for 50.5% of the total variance contribution, presented weakly for the geographical distinction (Figure 5b). However, regarding wines, the first two PCs explained 44.6% and 23.7% of the variance, comprising 68.3% of the total variance, revealing the most relevant for the geographical distinction. The first PC was strongly associated with the values of Sr, Mg, Li, and Na, whereas Mn, Rb, Cs, and Cd were the dominant variables in the second PC (Figure 5c).

In summary, PCA can be carried out to completely distinguish the wine of six regions and partially differentiate the soil, whereas it is not good for the discriminant of grape samples.

In addition, too many characteristic elements obtained from PCA increase the cost and reduce the efficiency of origin traceability. Therefore, the correlation analysis combined with LDA was employed to obtain fewer elements for origin recognition, which is also convenient for practical application. 

### 3.5. Discriminant Analysis

It was concluded that K and Li had significant correlations between not only soils and grape berries, but also between grape berries and wines (Figure 2 and Figure 3). Moreover, the BCF values of elements Sr and Li are closely related to the environment factors (Figure 4). Thus, it was hypothesized that K, Li, and Sr were the characteristic elements that could be implemented to distinguish the six regions in this study. To approve the hypothesis, three elements (K, Li, and Sr) were chosen as input parameters of linear discriminant analysis (LDA) to identify the geographical origin of soils, grapes, and wines.

For soil samples, the first discriminant function (F1) was plotted against the second one (F2) as shown in Figure 6a. The LDA analysis allowed us to distinguish among six regions with 93.3% certainty and a 90% cross-validation rate. Li and Sr showed the highest correlation in F1 and F2, respectively. As shown in Figure 7c, Li divided the six regions into four groups: BLY region, HY region, YL and XX regions, as well as MQ and WH regions. Moreover, Sr differentiated between YL and XX, and MQ and WH regions, respectively (Figure 7b).

The LDA of the grape data sets could be classified among the six studied regions with 96.7% certainty and a cross-validation rate based on Sr, Li, and K (Figure 6b). Li and Sr showed the highest correlation in F1 and F2, respectively, demonstrating that they were assumed as crucial variables to distinguish amongst grape samples from the six studied regions. Li distinguished the two regions (BLY and WH) from other regions (Figure 7f), whereas Sr distinguished XX and WH from the other four regions (Figure 7e).

The LDA analysis was also applied to the wine samples, with a 100% validation and cross-validation rate, indicating that Sr, Li, and K could be completely discriminate the wine samples’ geographical origins (Figure 6c). The variables of K showed the highest correlation in F2 to the classification amongst wine from the studied regions. As shown in Figure 7g, K differentiated all the studied regions. 

## 4. Discussion

### 4.1. Geographical Elemental Profile in Soils, Grapes, and Wines

In this study, the elemental concentrations of soil were in similar magnitude as those found by other researchers, with higher levels found in the soils of northern Italy and lower levels in Romania [18,26]. Zou et al. [28] found that Sr had relatively higher levels among trace elements in soils from other seven regions in China, which was consist with our research. These results could be explained by differences in parent rock. Moreover, previous studies had shown that the soils in YL, BLY, and XX were mainly sandy loam; the soils of YL and BLY were composed of Eum-Orthric Anthrosols, which were rich in trace elements [32,43,44]. Nevertheless, the soils in MQ and WH were primarily sandy soils, which were generally lacking in trace elements [45,46]. The soils with intermediate concentrations in HY may be related to the gravel and sandy soil texture [47]. There findings are consistent with ours, in that three groups of soils were classified based on the distinctive elemental profiles. Group 1 includes the soils with high concentrations of the majority elements in YL, BLY, and XX regions; group 3 comprises soils low in most elements in MQ and WH regions; group 2 contains soils in the HY region with intermediate concentrations between group 1 and group 3. These results demonstrate that the content of elements in soils are affected by different soil types and could be used for geographical identification [48]. In addition, no soil samples in this study exceeded the maximum permissible concentration according to Chinese soil environmental quality (Cu < 200 mg/kg, Zn < 300 mg/kg, and Cd < 0.6 mg/kg) [49].

Concerning grape samples, the concentration pattern of major elements in “Meili” grapes was the same with our previous study [31] in “Ecolly” grapes (K > Ca > Mg > Na), but the ranges in “Meili” grapes (3.2–3000 mg/kg) were lower than these in “Ecolly” grapes (8.96–4614 mg/kg) from the same regions. Pepi et al. [50] also reported the same ranges of major elements, but lower concentrations (3.77–1688.80 mg/kg) in “Glera” grapes from the Italian region. Moreover, Gao et al. [31] reported that minor and trace elements in “Ecolly” grapes decrease in the order of Sr > Rb > Mn > Ba > Li > Cs > Cd, which was almost the same pattern with this study (Rb > Sr > Mn > Ba > Li > Cs > Cd), except for Rb and Sr, and higher concentrations of these elements were found in “Ecolly” grapes. Protano et al. [51] observed a lower Sr (mean values 2773 μg/kg) in Sangiovese grapes from Tuscany. These findings implied that the uptake of elements could be affected by grape varieties [52]. Further, these results indicate that the element content of different varieties vary greatly, and could distinguish not only the origins, but also the grape varieties.

Regarding the concentrations of major elements in wines in this study, the mean concentrations of major elements ranging from 5.8 to 4000 mg/L. K were the most concentrated elements, with a mean range of 1900–4000 mg/L, followed by Mg (150–250 mg/L), Ca (72–120 mg/L), and Na (5.8–20). These findings are consistent with the patterns reported in our previous research in the same regions, with significantly lower content of K (885–2708 kg/L) and Ca (29–107 mg/L), higher levels of Mg (179–285 mg/L) and Na (6.0–26 mg/L) in “Ecolly” wine [31]. These findings may be caused by the differences in grape variety and in the winemaking process. Our study also presents similar elemental patterns to many other research studies from other regions of the world, with lower mean content ranges found from the regions, i.e., Spain (12.76–688.60 mg/L), Italy (7.32–597.14 mg/L), and Greece (2.94–1201.17 mg/L) [11,21,50]. Moreover, minor and trace elements of “Ecolly” wine in our previous study were the same order (Fe > Rb > Sr > Zn > Mn > Al > Cu > Ba > Li > Cs > Cd) as this study [31]. Hence, the mean concentration of each element in wine exhibited a wide range of variation, which was related to the geographical origin of grape varieties used or the winemaking process.

### 4.2. Correlation among the Soils, Grape Berries, and Wines

The transfer of the elements from the soil to the grape is complicated and is mainly determined by the lithological characteristics of the parent rock and geochemical outlines of the cultivation region; it is coupled with the ability of grapevines to take up the elements and translocate them to leaves and berries [51]. In contrast, a similarity in elemental profiles may well be found between the grapes and wines due to grape chemistry, together with various winemaking techniques [20,23]. 

Therefore, the influence of elemental concentrations in vineyard soils on the elemental composition of “Meili” grape berries was determined by the Pearson correlation coefficient (r). Only concentrations of K, Li, and Cd that presented significant correlations between soils and grape berries demonstrated that the concentrations of the elements in grape berries might be strictly controlled by the grapevine rather than by their concentrations in soil [51]. The BCF provides information about the relative availability of elements in soil for uptake in plant tissues [53]. In this study, eight elements (Mg, K, Ca, Cu, Zn, Rb, Sr, and Ba) presented the range of BCF values >0.01, the other seven elements showed the range of BCF values <0.01. These data suggest a higher absorption of these eight elements found in “Meili” grapes. In addition, the concentrations of Cd (mean values from 0.22 to 1.70 μg/kg) in grapes did not exceed the maximum permissible concentration (<50 μg/kg) according to the Chinese Food Safety Standard [54].

The elemental composition of wine is mainly influenced by the absorption of elements from vineyard soil by vine plants and the winemaking process [20,23]. The concentrations of six elements (Na, Mg, K, Zn, Li, and Cs) increased slightly after fermentation (TF > 1), probably as a result of extraction from the skin and seeds [55,56]. Additionally, the element concentrations of K, Mn, Rb, Sr, Li, and Cs present a significant positive correlation between the wines and the grape berries in this study (*p* < 0.05). These results indicated that, while vinification could affect the elemental composition of wine, the primary factor affecting the elemental composition of wine was likely geographical origin [23,56].

### 4.3. The Geographical Characteristic Elements K, Sr, and Li

The main factors that control the elements uptake by plants from the soil are geochemical, climatic, biological, and anthropogenic origins [57]. Therefore, elemental profiling could be a reliable method to assess geographical origin due to the relationship between the elemental composition of soil and that in the wine via the grapes produced in a given soil [58]. K, Sr, and Li were observed not only to be significantly correlated among soils, grapes, and wines, but their accumulation from soil to grape also showed a significant correlation with environmental factors in this study. Moreover, LDA was employed to confirm that these three elements could be critical markers in discriminating the provenance of the six regions in China.

Sr has high geochemical mobility and is easily absorbed by all parts of the vine, including grapes, due to its chemical affinity to Ca [59]; it is then transferred into wine without being altered during the winemaking process [60]. Therefore, Sr in wine directly originates from the soil, and it could be used as a reference for the geographical origin of plants [19]. Moreover, numerous studies show promising perspectives for provenance determinations of wine through the elemental and isotopic composition of Sr [11,18,19,26,51,61]. Epova, E. N. et al. [19] found that comparing Sr concentrations in authentic and suspicious Bordeaux wines could reveal significant divergence, so that Sr could be regarded as an origin tracer. It is also consistent with our study that the concentrations of Sr vary among the studied regions. Additionally, the TF values of Sr in the grape–wine system were below 1 (Appendix A), further indicating that Sr was less affected by winemaking processes [60]. In previously studies, Sr appeared the most frequently among all of the elemental analyses for wine origin identification [29]. In this study, the BCF values of Sr showed a significant negative correlation with average relative humidity, reflecting that the transfer of Sr from soil to grape berries was related closely to climatic conditions [19]. The results demonstrate that the content of Sr could be applied for the discrimination of regions in China.

Li is neither an essential factor nor a micronutrient for the vine, but it still plays a vital role in the human body as an effective mood stabilizer [62]. Commonly, the elemental and isotopic Li responds to silicate weathering intensity and can be a weathering tracer [63]. Weathering processes can affect the total and bioavailable concentrations of chemical species in soil [51]. Fan et al. [29] found that element Li appeared eleven times, second only to Sr for origin tracing in many previous research studies. In our study, Li showed a significant negative correlation between soil and grape berries, suggesting that a higher concentration of Li may inhibit the accumulation of Li in grape berries. This result is similar to previous research by Ren et al., who found that the Li accumulation of *Apocynum* decreased when the lithium concentration increased to 400 mg/kg [64]. Li presented a significant positive correlation between grape berry and wine that could be explained with the result of its presence in the vineyard’s soil and its uptake by the vine plant [23]. Additionally, the BCF value of Li has a significant negative correlation with temperature and altitude, highlighting that the geographical and climatic factors predominantly influence the uptake of Li by grapes from the soil. Moreover, Su et al. [30] classified 142 wines from five Chinese regions by Li and the other eight elements. The wines from Shandong in China were effectively distinguished from France by Li and the other seven inorganic elements [65]. These results confirm that Li could be an essential element for identifying the geographical origin of wine in China.

The major element, K, is essential for a vine plant’s health and it accumulates in grape berries naturally during the growing season; however, it is also added during the winemaking process through potassium metabisulphite [20,58]. In this study, K was significantly different in soil, grape, and wine among the six regions, suggesting the possibility to use it to identify the geographical origin. K was the most abundant element in grape berries and wines, and was second only to Fe in the studied soil, which demonstrate K is the predominant cation in vinification [12]. Pearson correlation analysis found that K showed significant positive correlations among the soils, grape berries, and wines, highlighting that it is crucial as a variable related to vineyard soil and grape maturity. Furthermore, previous document reported K was one of the important parameters for wine region classification [29,31]. Consequently, the elements that could establish a reliable correspondence among soils, grapes, and wines in different Chinese regions are K, Sr, and Li.

## 5. Conclusions

Diverse patterns of fifteen elements in soils, grapes, and wines from six regions in China were observed by a combination of ICP-MS and chemometrics. BCF and TF values demonstrated the migration of the elements from soil to grape and wine. This is the first study to evaluate the relationships of elements in the soil–grape–wine system and BCF-environmental factor system in China. K and Li present significant correlations among soils, grapes, and wines, as do Sr and Li between BCF values and environmental factors. Furthermore, LDA combined with the Pearson correlation analysis exhibited excellent capacity in the authenticity of soil, grape, and wine regions. Three elements, K, Sr, and Li, show potential as tracers for geographical origin identification of Chinese wine. Nevertheless, more samples from different regions, grape varieties, and vintages should be analyzed in order to consolidate this conclusion.

## Figures and Tables

**Figure 1 foods-10-03108-f001:**
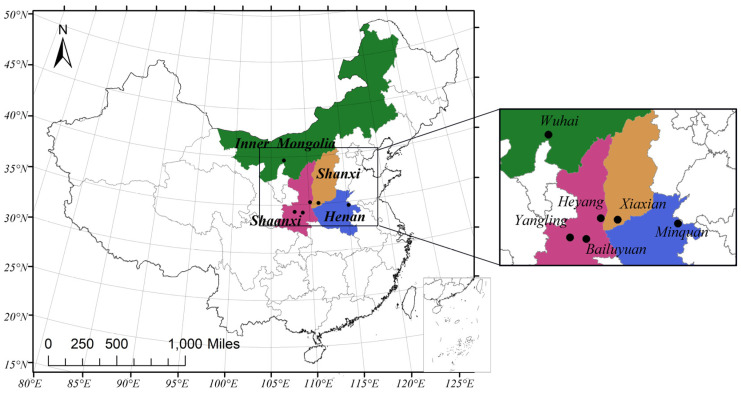
Map of six regions where soil, grape berries, and wine samples were collected: YL (Yangling, Shaanxi), BLY (Bailuyuan, Shaanxi), HY (Heyang, Shaanxi), XX (Xiaxian, Shanxi), MQ (Minquan, Henan), and WH (Wuhai, Inner Mongolia autonomous region).

**Figure 2 foods-10-03108-f002:**
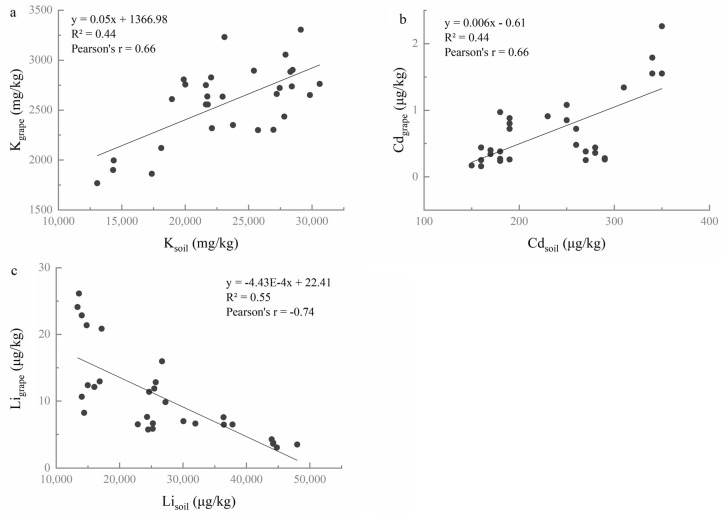
The relationships among the elements K (**a**), Cd (**b**), and Li (**c**) between soils and grapes.

**Figure 3 foods-10-03108-f003:**
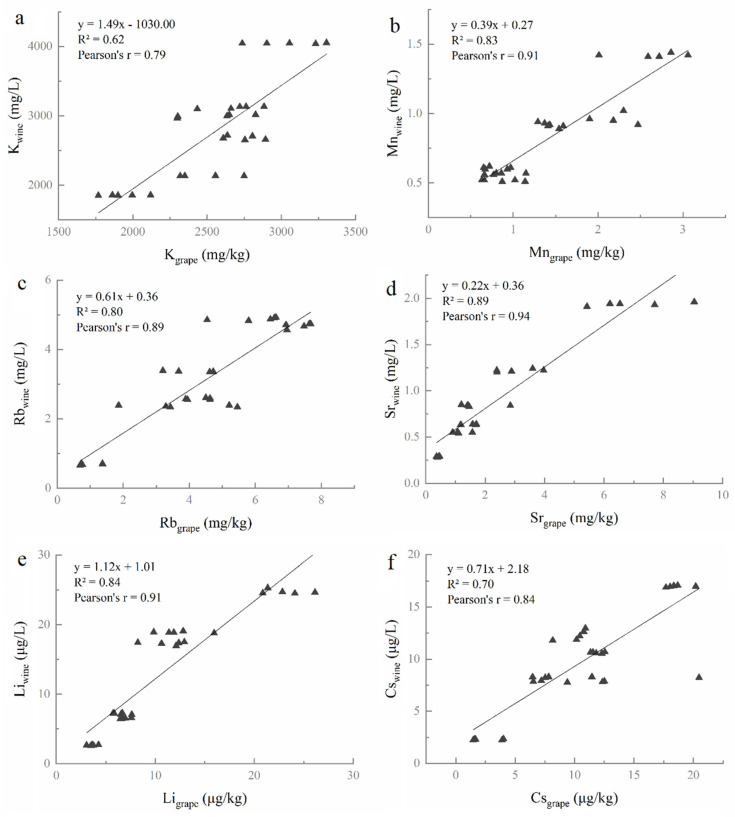
The relationships among the elements K (**a**), Mn (**b**), Rb (**c**), Sr (**d**), Li (**e**), and Cs (**f**) between grape berries and wines.

**Figure 4 foods-10-03108-f004:**
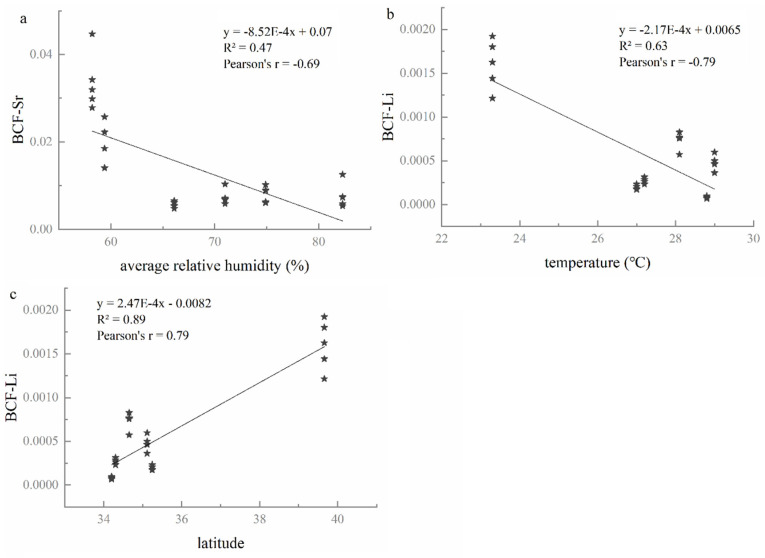
The relationships the BCF values and environmental factors: (**a**): correlation between BCF value of Sr and average relative humidity; (**b**): correlation between BCF value of Li and temperature; (**c**): correlation between BCF value of Li and latitude.

**Figure 5 foods-10-03108-f005:**
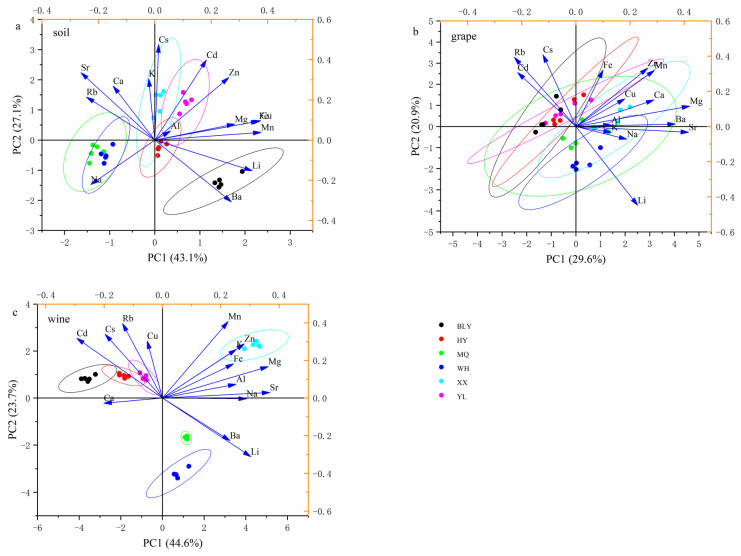
Score and loading plots of PCA for soil (**a**), grape (**b**), and wine (**c**): BLY = Bailuyuan; HY = Heyang; MQ = Minquan; WH = Wuhai; XX = Xiaxian; YL = Yangling.

**Figure 6 foods-10-03108-f006:**
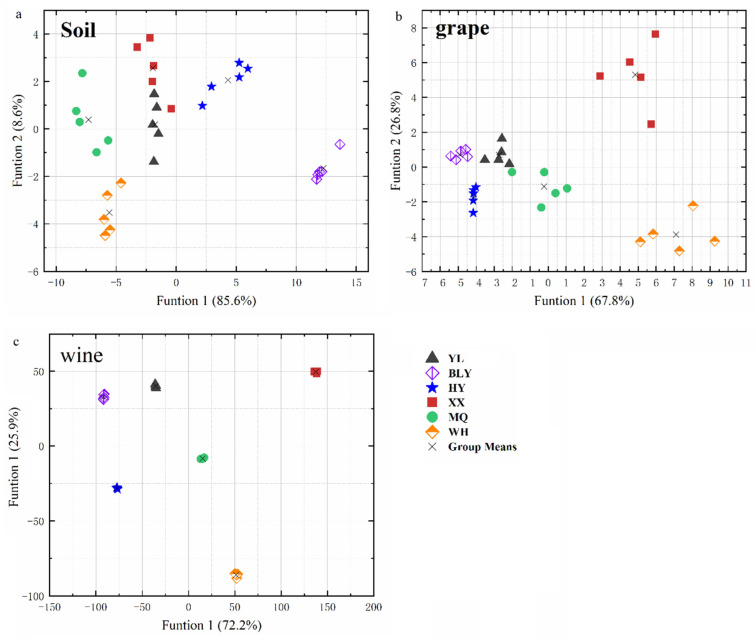
Linear discriminant analysis (LDA) performed on data expressed as concentrations of the three elements (K, Sr, and Li) in soil (**a**), grape (**b**), and wine samples (**c**) from the six studied regions: YL = Yangling; BLY = Bailuyuan; HY = Heyang; XX = Xiaxian; MQ = Minquan; WH = Wuhai.

**Figure 7 foods-10-03108-f007:**
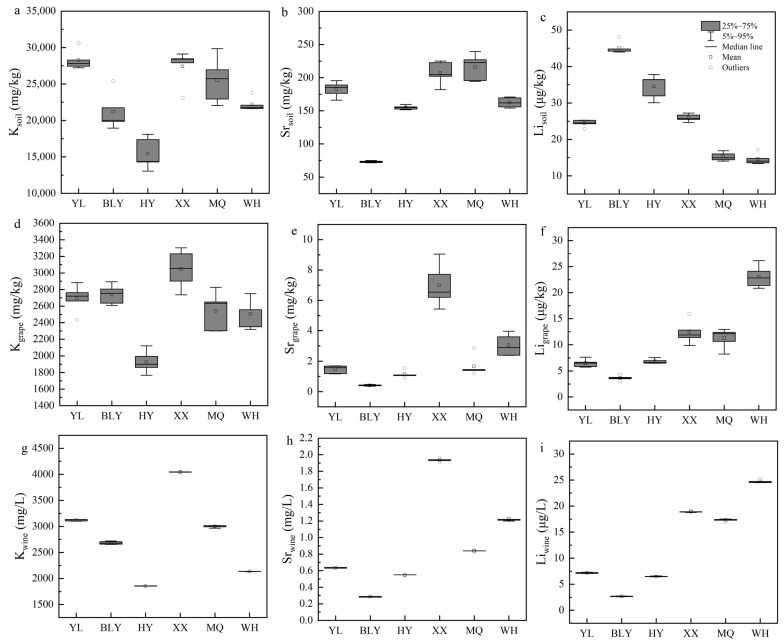
Box plots with elemental contents of K, Sr, and Li in soils (**a**–**c**), grape berries (**d**–**f**), and wines (**g**–**i**) from the six regions: YL = Yangling; BLY = Bailuyuan; HY = Heyang; XX = Xiaxian; MQ = Minquan; WH = Wuhai.

**Table 1 foods-10-03108-t001:** Information on samples, locations, and weather conditions in July and August of the six studied regions in China.

Regions	Latitude	Longitude	Altitude (m)	Precipitation (mm)	Average Temperature (°C)	Average Relative Humidity (%)
Yangling, Shaanxi (YL)	34.3	108.1	521	262.7	27.2	74.9
Bailuyuan, Shaanxi (BLY)	34.2	109.1	713	185.8	28.8	66.1
Heyang, Shaanxi (HY)	35.2	110.2	716	158	27	71
Xia county, Shanxi (XX)	35.1	111.2	402.9	66.6	29	58.2
Minquan, Henan (MQ)	34.7	115.2	60.6	538.8	28.1	82.3
Wuhai, Inner Mongolia (WH)	39.7	106.8	1105.6	171.1	23.3	59.4

Note: information obtained from National Meteorological Information Center of China.

**Table 2 foods-10-03108-t002:** Means and standard deviations of measured elements corresponding to soils according to geographical origins.

Element	YL	BLY	HY	XX	MQ	WH
Fe	330,000 ± 1000 ^a^	36,000 ± 5600 ^a^	17,000 ± 2100 ^c^	25,000 ± 3300 ^b^	8800 ± 2300 ^d^	9300 ± 1600 ^d^
K	28,000 ± 1400 ^a^	21,000 ± 2600 ^b^	15,000 ± 2200 ^c^	27,000 ± 2400 ^a^	26,000 ± 3100 ^a^	22,000 ± 900 ^b^
Ca	22,000 ± 1200 ^a^	17,000 ± 2700 ^b^	23,000 ± 3500 ^a^	26,000 ± 5200 ^a^	23,000 ± 2800 ^a^	23,000 ± 3800 ^a^
Al	12,000 ± 6200 ^c^	19,000 ± 3000 ^ab^	5400 ± 1900 ^d^	20,000 ± 3600 ^a^	15,000 ± 1500 ^bc^	14,000 ± 1400 ^c^
Na	5300 ± 200 ^d^	6600 ± 800 ^b^	6400 ± 200 ^bc^	5900 ± 300 ^c^	9200 ± 500 ^a^	6700 ± 300 ^b^
Mg	4700 ± 500 ^b^	5800 ± 1400 ^a^	3000 ± 400 ^c^	5100 ± 600 ^ab^	3200 ± 400 ^c^	3400 ± 400 ^c^
Mn	700 ± 29 ^b^	800 ± 140 ^a^	540 ± 18 ^c^	560 ± 17 ^c^	360 ± 38 ^d^	330 ± 20 ^d^
Sr	180 ± 11 ^b^	73 ± 1.4 ^d^	160 ± 3.2 ^c^	210 ± 18 ^a^	220 ± 20 ^a^	160 ± 7.5 ^c^
Zn	80 ± 3.0 ^a^	62 ± 0.9 ^c^	73 ± 2.4 ^b^	72 ± 5.7 ^b^	48 ± 3.2 ^d^	46 ± 3.2 ^d^
Rb	32 ± 17 ^b^	6.4 ± 1.4 ^c^	10 ± 3.6 ^c^	65 ± 33 ^a^	57 ± 3.4 ^a^	57 ± 5.9 ^a^
Li	24 ± 1.0 ^c^	45 ± 1.7 ^a^	34 ± 3.3 ^b^	26 ± 1.0 ^c^	15 ± 1.2 ^d^	15 ± 1.5 ^d^
Cu	29 ± 1.3 ^a^	31 ± 0.9 ^a^	25 ± 1.4 ^b^	25 ± 2.5 ^b^	13 ± 2.5 ^c^	12 ± 1.2 ^c^
Ba	6.1 ± 1.1 ^b^	330 ± 12 ^a^	8.7 ± 1.3 ^b^	9.2 ± 4.8 ^b^	8.2 ± 0.6 ^b^	7.1 ± 0.9 ^b^
Cs	6.6 ± 0.4 ^a^	1.2 ± 0.3 ^d^	4.0 ± 0.7 ^b^	7.2 ± 1.8 ^a^	2.9 ± 0.4 ^c^	3.1 ± 0.4 ^bc^
Cd	0.34 ± 0.02 ^a^	0.20 ± 0.02 ^d^	0.26 ± 0.01 ^c^	0.28 ± 0.01 ^b^	0.17 ± 0.01 ^e^	0.17 ± 0.02 ^e^

Note: five soil samples were collected from each region and analyzed in triplicate. The elemental concentrations of the soil samples are expressed in mg/kg; YL = Yangling; BLY = Bailuyuan; HY = Heyang; XX = Xiaxian; MQ = Minquan; WH = Wuhai. Different letters in the same row indicate statistically significant difference (*p* < 0.05).

**Table 3 foods-10-03108-t003:** Means and standard deviations of measured elements corresponding to grape berries according to geographical origins.

Element	YL	BLY	HY	XX	MQ	WH
K	2700 ± 170 ^b^	2700 ± 120 ^b^	1900 ± 130 ^c^	3000 ± 230 ^a^	2500 ± 230 ^b^	2500 ± 180 ^b^
Ca	240 ± 93 ^a^	150 ± 42 ^a^	240 ± 83 ^a^	220 ± 53 ^a^	220 ± 85 ^a^	23 ± 62 ^a^
Mg	120 ± 41 ^b^	110 ± 16 ^b^	110 ± 25 ^b^	180 ± 18 ^a^	120 ± 28 ^b^	130 ± 22 ^b^
Na	3.3 ± 0.3 ^c^	3.2 ± 0.5 ^c^	3.8 ± 0.5 ^c^	7.8 ± 2.0 ^b^	13 ± 1.4 ^a^	3.6 ± 0.3 ^c^
Fe	6.9 ± 1.8 ^a^	6.3 ± 4.5 ^a^	6.7 ± 1.4 ^a^	6.3 ± 0.6 ^a^	6.2 ± 1.0 ^a^	5.2 ± 0.9 ^a^
Rb	6.0 ± 0.9 ^b^	7.3 ± 0.4 ^a^	4.3 ± 0.4 ^c^	4.2 ± 0.7 ^c^	3.9 ± 1.5 ^c^	1.0 ± 0.4 ^d^
Cu	2.6 ± 0.2 ^ab^	1.9 ± 0.3 ^b^	3.3 ± 3.2 ^ab^	4.5 ± 2.7 ^a^	2.5 ± 0.8 ^ab^	1.7 ± 0.3 ^b^
Sr	1.5 ± 0.3 ^c^	0.4 ± 0.04 ^d^	1.2 ± 0.2 ^cd^	7.0 ± 1.4 ^a^	1.7 ± 0.7 ^c^	3.0 ± 0.7 ^b^
Mn	1.8 ± 0.4 ^b^	0.9 ± 0.2 ^c^	1.7 ± 0.5 ^b^	2.7 ± 0.4 ^a^	0.8 ± 0.2 ^c^	0.9 ± 0.2 ^c^
Zn	1.3 ± 0.2 ^b^	1.5 ± 0.5 ^ab^	1.5 ± 0.4 ^ab^	1.8 ± 0.4 ^a^	1.4 ± 0.2 ^ab^	1.2 ± 0.3 ^b^
Al	1.2 ± 0.1 ^ab^	1.1 ± 0.2 ^b^	1.7 ± 0.6 ^a^	1.6 ± 0.3 ^a^	1.5 ± 0.1 ^ab^	1.3 ± 0.5 ^ab^
Ba	300 ± 120 ^ab^	100 ± 22 ^c^	180 ± 31 ^bc^	380 ± 110 ^a^	380 ± 180 ^a^	260 ± 23 ^ab^
Li	6.5 ± 0.8 ^c^	3.6 ± 0.4 ^d^	6.8 ± 0.5 ^c^	12 ± 2.3 ^b^	11 ± 1.9 ^b^	23 ± 2.1 ^a^
Cs	10 ± 1.1 ^b^	12 ± 0.5 ^b^	19 ± 1.0 ^a^	9.6 ± 2.8 ^b^	11 ± 5.8 ^b^	2.5 ± 1.3 ^c^
Cd	1.7 ± 0.4 ^a^	0.9 ± 0.1 ^b^	0.7 ± 0.3 ^b^	0.3 ± 0.1 ^c^	0.4 ± 0.1 ^c^	0.2 ± 0.0 ^c^

Note: five grape samples were collected from each region and analyzed in triplicate. The elemental concentrations of the grape berries are expressed in mg/kg from K to Al and in μg/kg from Ba to Cd, respectively; YL = Yangling; BLY = Bailuyuan; HY = Heyang; XX = Xiaxian; MQ = Minquan; WH = Wuhai. Different letters in the same row indicate statistically significant difference (*p* < 0.05).

**Table 4 foods-10-03108-t004:** Means and standard deviations of measured elements corresponding to wines according to geographical origins.

Element	YL	BLY	HY	XX	MQ	WH
K	3100 ± 16 ^b^	2700 ± 28 ^d^	1900 ± 3.0 ^f^	4000 ± 5.3 ^a^	3000 ± 21 ^c^	2100 ± 2.4 ^e^
Mg	180 ± 1.9 ^c^	150 ± 0.9 ^f^	170 ± 0.8 ^e^	250 ± 2.2 ^a^	200 ± 0.9 ^b^	180 ± 0.5 ^d^
Ca	120 ± 1.0 ^a^	92 ± 1.7 ^c^	120 ± 16 ^ab^	72 ± 1.4 ^d^	110 ± 0.4 ^b^	88 ± 0.2 ^c^
Na	6.2 ± 0.0 ^e^	6.9 ± 0.1 ^d^	5.8 ± 0.0 ^f^	18 ± 0.2 ^b^	20 ± 0.1 ^a^	7.4 ± 0.1 ^c^
Rb	4.9 ± 0.0 ^a^	4.7 ± 0.1 ^b^	2.6 ± 0.0 ^d^	3.4 ± 0.0 ^c^	2.4 ± 0.0 ^e^	0.7 ± 0.0 ^f^
Fe	2.0 ± 0.0 ^b^	1.1 ± 0.3 ^d^	2.0 ± 0.0 ^b^	2.2 ± 0.2 ^a^	1.6 ± 0.0 ^c^	1.6 ± 0.0 ^c^
Zn	1.4 ± 0.0 ^c^	0.71 ± 0.01 ^f^	1.5 ± 0.0 ^b^	2.1 ± 0.1 ^a^	1.3 ± 0.0 ^d^	0.98 ± 0.01 ^e^
Sr	0.63 ± 0.01 ^d^	0.29 ± 0.01 ^f^	0.55 ± 0.00 ^e^	1.9 ± 0.02 ^a^	0.84 ± 0.01 ^c^	1.2 ± 0.01 ^b^
Mn	0.95 ± 0.05 ^b^	0.57 ± 0.01 ^e^	0.92 ± 0.01 ^c^	1.4 ± 0.01 ^a^	0.61 ± 0.01 ^d^	0.52 ± 0.01 ^f^
Al	0.36 ± 0.01 ^b^	0.38 ± 0.09 ^b^	0.45 ± 0.05 ^b^	0.73 ± 0.27 ^a^	0.48 ± 0.01 ^b^	0.52 ± 0.22 ^b^
Cu	0.22 ± 0.00 ^e^	0.35 ± 0.01 ^a^	0.27 ± 0.01 ^c^	0.32 ± 0.01 ^b^	0.21 ± 0.00 ^f^	0.24 ± 0.01 ^d^
Ba	120 ± 0.6 ^b^	70 ± 1.0 ^f^	85 ± 0.4 ^e^	110 ± 0.2 ^d^	150 ± 1.0 ^a^	110 ± 0.9 ^c^
Li	7.2 ± 0.1 ^d^	2.7 ± 0.1 ^f^	6.5 ± 0.1 ^e^	19 ± 0.1 ^b^	17 ± 0.2 ^c^	25 ± 0.3 ^a^
Cs	12 ± 0.5 ^b^	11 ± 0.1 ^c^	17 ± 0.1 ^a^	7.8 ± 0.1 ^e^	8.3 ± 0.0 ^d^	2.3 ± 0.0 ^f^
Cd	0.36 ± 0.05 ^c^	0.56 ± 0.03 ^a^	0.45 ± 0.03 ^b^	0.56 ± 0.03 ^a^	0.22 ± 0.02 ^d^	0.14 ± 0.03 ^e^

Note: five wine samples were collected from each region and analyzed in triplicate. The elemental concentrations of the wine samples are expressed in mg/L from K to Cu and in μg/L from Ba to Cd, respectively; YL = Yangling; BLY = Bailuyuan; HY = Heyang; XX = Xiaxian; MQ = Minquan; WH = Wuhai. Different letters in the same row indicate statistically significant difference (*p* < 0.05).

## Data Availability

Data is contained within the article or references.

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
