# Peer review of "From Soil to Grape and Wine: Geographical Variations in Elemental Profiles in Different Chinese Regions"

_foods, 2021, doi:10.3390/foods10123108_

Round 1
Reviewer 1 Report
The Contribution titled From soil to grape and wine: the geographical variation of elemental profile in China is focused on comparisons among elements in soils, grape berries, and wines from six regions. The relationships between BCF and environmental factors (temperature, humidity, etc.) were investigated to obtain the mineral elemental profiles that close relative to the geographical origins
From the point of view of viticultural practice, the topic brings current knowledge. The paper is prepared at a good level, however, I would recommend the authors to complete the discussions, especially in the part focused on Li, Sr and K, which is insufficiently commented. In particular, I consider the conclusion to be insufficient, which should emphasize the main benefits of the results for production practice. Several errors also contain a list of References.
Reviewer 2 Report
This study might be interesting, but the analytical methods have not carefully assessed and validated, hence the results proposed are likely affected by systematic errors. Also the quality of the data presentation (both as tables as well as in figures) is far to be acceptable. Univariate correlations must be replaced by a much more informative multivariate approach (e.g. PCA). Specific comments, remarks and suggestions are reported as sticky notes in the enclosed files.

Round 2
Reviewer 2 Report
The authors have in part kept into consideration my suggestions. However, at least two otf them have not conveniently addressed.
1) Data in the tables are still not harmonized according uncertainty. Some examples: Table 2, Cs, 6.6±0.4 and NOT 6.64±0.44; Al, 12000±6000 and NOT 11600±6200...and so on. Please correct in this way throughout the paper.
2) PCA is necessary in this paper. Variability among concentrations of different matrices/elements can be overcome by centering/autoscaling data.
